

# Effects of trapping effort and trap placement on estimating abundance of Humboldt's flying squirrels

Matthew J. Weldy[1,2], Todd M. Wilson[2], Damon B. Lesmeister[1,2] and Clinton W. Epps[1]

[1] Department of Fisheries and Wildlife, Oregon State University, Corvallis, OR, United States of America
[2] Pacific Northwest Research Station, U.S.D.A. Forest Service, Corvallis, OR, United States of America

## ABSTRACT

Live trapping is a common tool used to assess demography of small mammals. However, live-trapping is often expensive and stressful to captured individuals. Thus, assessing the relative tradeoffs among study goals, project expenses, and animal well-being is necessary. Here, we evaluated how apparent bias and precision of estimates for apparent annual survival, abundance, capture probability, and recapture probability of Humboldt's flying squirrels (*Glaucomys oregonensis*) varied with the number of secondary trapping occasions. We used data from forested sites trapped on 12 consecutive occasions annually in the HJ Andrews Experimental Forest (9 sites, 6 years) and the Siuslaw National Forest (seven sites, three years) in Oregon. We used Huggins robust design models to estimate parameters of interest for the first 4, 8, and 12 trapping occasions. We also estimated the effect of attaching Tomahawk traps to tree boles on site- and year-specific flying squirrel capture frequencies. Our estimates with 12 occasions were similar to those from previous studies. Abundances and capture probabilities were variable among years on both sites; however, variation was much lower on the Siuslaw sites. Reducing the length of primary trapping occasions from 12 to 8 nights had very little impact on parameter estimates, but further reducing the length of primary trapping occasions to four nights caused substantial apparent bias in parameter estimates and decreased precision. We found that attaching Tomahawk traps to tree boles increased the site- and year-specific capture frequency of flying squirrels. Our results suggest that live-trapping studies targeting Humboldt's flying squirrels in the Pacific Northwest of the United States could reduce per-site costs and stress to captured individuals without biasing estimates by reducing the length of primary trapping occasions to 8 nights. We encourage similar analyses for other commonly-trapped species in these and other ecosystems.

Corresponding author
Matthew J. Weldy,
Matthew.Weldy@Oregonstate.edu

# INTRODUCTION

Conservation and management of small mammals commonly require accurate estimates of abundance and vital rates (*Williams, Nichols & Conroy, 2002*). These metrics are often assessed using capture-recapture data obtained from live-trapping studies, where animals

are captured, uniquely marked, and released back into the population to be recaptured. Live trapping remains an important technique in these studies but can be both labor intensive and physically demanding, and project costs are sensitive to the number of project trap nights. In addition, capture events can cause considerable stress or even mortality of target and non-target species, especially during longer trapping sessions (*Sikes, 2016*), although this pattern has been debated for some species (*Rosenberg & Anthony, 1993*). To minimize invasiveness of vertebrate research, it is important to carefully consider how best to incorporate the principles emphasized by the three R's (Replace, Reduce, Refine) of animal welfare, while still obtaining sufficient samples from which to draw inferences from the data (*Russell & Burch, 1959*; *Powell & Proulx, 2003*; *Villette et al., 2016*).

Numerous analytical methods have been applied to live-trapping data. The primary divide among these methods is the consideration of imperfect detection probabilities, which occurs when an individual or species is present but not detected. Methods that do not account for imperfect detection report unadjusted estimates such as the minimum number of known individuals alive (e.g., *Coppeto et al., 2006*; *Fauteux et al., 2012*), which have lower data requirements (*Banks-Leite et al., 2014*). However, failing to account for variation in detection probability when detection probabilities are less than one can cause substantial bias in the estimation of important demographic parameters (*Nichols & Pollock, 1983*; *Kéry & Schmidt, 2008*; *Williams, Nichols & Conroy, 2002*; *Kellner, Urban & Swihart, 2013*). For example, after accounting for variable detection probabilities among species monitored during the Swiss breeding bird survey, *Kéry & Schmidt (2008)* reported substantial differences and underestimation among observed and estimated species-specific distributions.

Abundance of small mammals has been of research interest in the Pacific Northwest USA (hereafter, PNW) because small mammals comprise a large proportion of the prey base for many avian and mammalian predators, including mustelids and owls. Two species of flying squirrels, the recently describe Humboldt's flying squirrel (*Glaucomys oregonensis*; *Arbogast et al., 2017*) and the northern flying squirrel (*G. sabrinus*), have been of interest because they serve as prey for the federally threatened northern spotted owl (*Strix occidentalis caurina*; *Forsman, Meslow & Wight, 1984*; *Forsman et al., 2004*; *USFWS, 1990*). Flying squirrels also contribute to the maintenance of ecosystem health through the dispersal of hypogeous fungi, berries, and seeds (*Maser, Trappe & Nussbaum, 1978*; *Bowers & Dooley Jr, 1993*; *Carey et al., 1999*; *Smith, 2007*). Much of the early research on flying squirrels focused on differences in abundance among young, mature, and old-growth forests (*Rosenberg & Anthony, 1992*; *Carey et al., 1999*; *Holloway & Smith, 2011*). More recent research emphasis has been on understanding the effects of timber harvest strategies implemented to speed the development of late-seral forest characteristics from young, managed forests (*Carey, 2000*; *Holloway et al., 2012*; *Manning, Hagar & McComb, 2012*; *Wilson & Forsman, 2013*).

*Carey, Biswell & Witt (1991)* developed a commonly used protocol for sampling flying squirrels and other arboreal rodents in PNW. This protocol recommends two weeks of trapping (four trap nights per week) and two traps per station: one trap attached to a tree at diameter breast height (~1.4 m above ground), and one trap placed on the ground. Some studies have increased the trapping period from two to three weeks (12 trap nights

total) for two reasons. First, several multi-year studies reported relatively low numbers of individuals captured the first time a stand is sampled, either with or without pre-baiting, relative to subsequent trapping sessions (e.g., *Carey, 2000*; *Lehmkuhl et al., 2006*; *Holloway et al., 2012*). Second, some studies reported fewer squirrel captures during the first week of study, and a subsequent increase of captures during the second week, especially in structurally complex forests (*Wilson, 2010*). Other trapping protocols have been used to sample arboreal rodents in the PNW (*Rosenberg & Anthony, 1993*; *Rosenberg, Overton & Anthony, 1995*; *Ransome & Sullivan, 1997*). However, we are unaware of any studies that examined how increasing or decreasing the number of trap nights influences flying squirrel abundance estimates. *Weldy et al. (2019)* recently estimated Humboldt's flying squirrel abundance using Huggins closed-population models and found the estimates of precision were very small, in some cases small enough that the reported estimates resemble a full census, suggesting that reducing trap nights could be considered. Similarly, considerable time and energy goes into hanging traps on trees. *Carey, Biswell & Witt (1991)* recommended using both ground and tree traps because they observed a strong but variable selection for tree traps by flying squirrels. Despite general evidence that trap placement can strongly affect animal captures (*Risch & Brady, 1996*; *Trolle & Kéry, 2005*), however, there is little empirical evidence that tree traps increase the frequency of flying squirrel captures in the PNW.

Our first objective for this study was to measure the apparent bias and precision of abundance, apparent annual survival, and capture and recapture probability estimates using one week (four trap nights), two weeks (eight trap nights), and three weeks (12 nights) of trapping. Our second objective was to estimate the relative effectiveness of traps placed on the ground compared to traps attached to tree boles. We predicted that two consecutive weeks of live trapping would yield reliable and precise estimates of flying squirrel abundance, apparent annual survival, and capture and recapture probabilities. We also predicted there would be no difference in capture frequencies between traps attached to tree boles and those placed on the ground.

## MATERIALS & METHODS

### Study areas

We used live-trapping data collected during two western Oregon studies (Fig. 1). The first study (hereafter SIU) consisted of seven sites in the Siuslaw National forest. SIU sites were located across the Oregon Coast Range. Four sites were in natural late-successional stands, and three were in managed forests. The SIU sites were dominated by Douglas-fir (*Pseudotsuga menziesii*) and western hemlock (*Tsuga heterophylla*). Average elevations on the SIU sites ranged from 830–1,040 m. The second study (hereafter HJA) consisted of nine sites located in the HJ Andrews Experimental Forest, part of the Willamette National Forest, on the western slope of the Oregon Cascade Range. The HJA sites were all located in a late-successional forest (>400 years old), and dominated by large (>81 cm diameter at breast height) Douglas-fir, western hemlock, and Pacific silver fir (*Abies amabilis*; *Cissel, Swanson & Weisberg, 1999*; *Schulze & Lienkaemper, 2015*). Average HJA site elevations ranged from
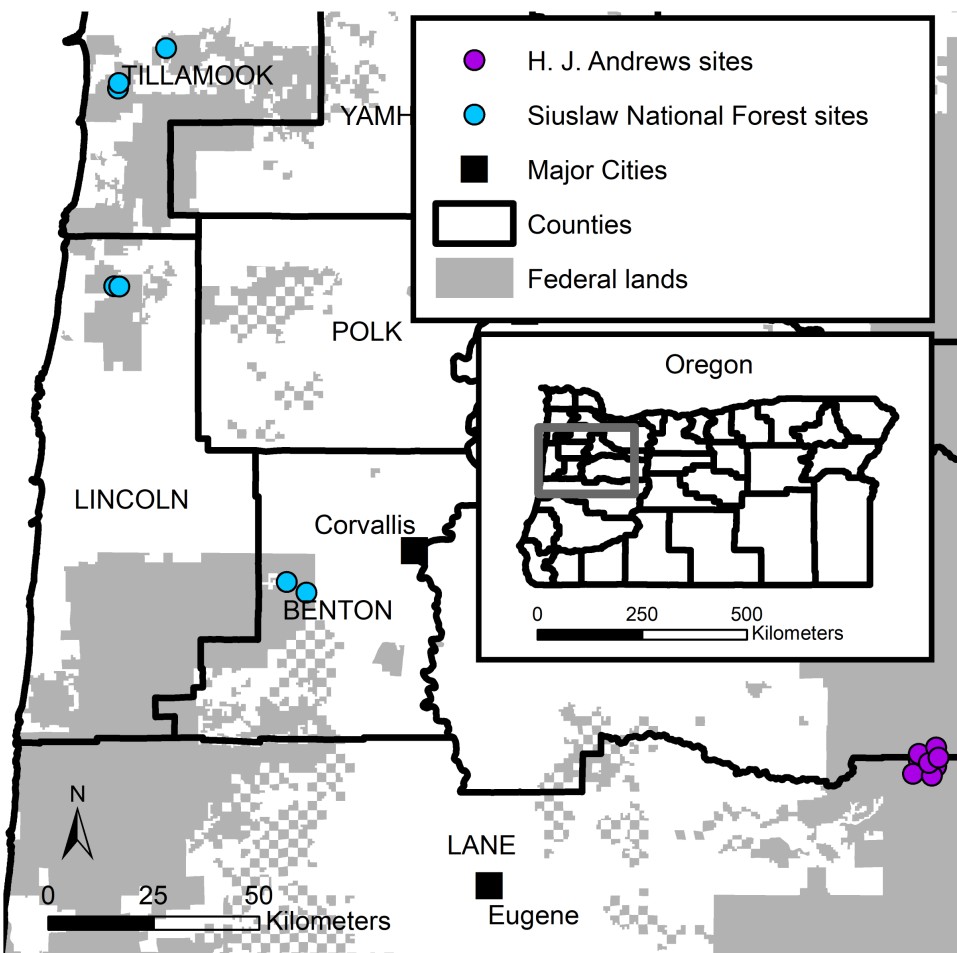

**Figure 1** Humboldt's flying squirrel trapping locations in the HJ Andrews Experimental (purple) Forest and the Siuslaw National Forest (blue) in western Oregon.

683–1,244 m. Weather for both studies was typically hot and dry May–September, and cool and wet October–April, with winter precipitation primarily consisting of rain at low elevations (<1,000 m) and snow at high elevations (>1,000 m; *Swanson & Jones, 2002*).

## Capture methods

Trapping methods on the SIU and HJA sites were similar. We established 64 trap stations arranged in an 8 × 8 array (7.84 ha) with 40 m (corrected for slope) between traps on each site. We placed two (128 traps per site) Tomahawk Model 201 live traps (Tomahawk Live Trap, WI, USA) at each trap station within 5 m of the trap station center. One trap was attached to a suitable tree bole (~1.4 m from the ground), the second trap was placed on the ground. On the HJA sites, we placed both traps on the ground at an average of 10.5% (SE = 0.7%; 95% CI [8.92%–12.03%]) of the trap stations due to lack of available trees. We placed traps near habitat features, such as fallen logs, to increase the likelihood of small mammals encountering traps (*Carey, Biswell & Witt, 1991*).

Live trapping occurred from 2014–2016 on the SIU sites and 2011–2016 on the HJA sites. Each site was trapped once annually for three consecutive weeks between September and early December. Each week consisted of four consecutive trap nights. We checked and reset traps once per day. To reduce trapping related mortalities, we covered each trap with a waxed cardboard carton and placed a cotton- or polyester- filled nest box inside each trap. On the HJA sites, we pre-baited each trap once, approximately 12 days before the trapping session. We did not prebait traps on the SIU sites. Bait consisted of a mixture of almond or peanut butter, molasses, oats, and sunflower seeds (HJA). Total trap nights were adjusted for sprung or otherwise unavailable traps (*Nelson & Clark, 1973*).

We tagged individuals with unique ear tags and recorded reproductive condition, species, sex, and body weight (g). We necropsied all trapping mortalities to validate field identification of species, sex, and reproductive condition. Field work for this project was collected under Oregon Department of Fish and Wildlife scientific take permits from 2011 to 2016 (STP #s 122-11, 109-12, 039-13, 052-14, 118-14, 047-15, 124-15, 058-16, and 094-16). This work was funded by the USDA Forest Service and conducted on Forest Service lands with approval from the Siuslaw National Forest, Willamette National Forest, and the HJ Andrews Forest research group. The HJA live-trapping protocols were approved by Oregon State University's Institutional Animal Care and Use Committee (ACUP: #4191 (2011–2013), #4590 (2014–2016)); and the SIU live-trapping protocols were approved by the USDA Forest Service Starkey Institute for Animal Care and Use Committee (USFS: # 92-F-0004). In addition, both live-trapping protocols were consistent with the American Society of Mammalogists guidelines for the use of wild mammals in research and education (*Sikes, 2016*).

## Sequential modelling procedure

We used Huggins robust design models implemented in RMark (*Laake, 2013*) to estimate apparent annual survival ($\varphi$), capture probability ($c$), recapture probability ($p$), temporary emigration ($\gamma''$), and temporary immigration ($\gamma'$) of flying squirrels. The Huggins robust design model structure is a combination of Huggins closed capture models and Cormack–Jolly–Seber live recapture models (*Kendall, Pollock & Brownie, 1995*; *Kendall & Nichols, 1995*; *Kendall, Nichols & Hines, 1997*). We chose to consider only the Huggins robust design model for two reasons: (1) considering other models could have introduced parameter estimate bias inherent to different model structures, and (2) it provided inference on multiple parameters commonly considered when analyzing small mammal mark-recapture datasets. Robust design data consists of primary occasions and secondary occasions recorded within primary occasions (*Pollock, 1982*). Capture and recapture probabilities were estimated within primary occasions with the Huggins component (*Huggins, 1989*; *Huggins, 1991*), and apparent annual survival, temporary emigration, and temporary immigration were estimated between primary occasions with the Cormack–Jolly–Seber component (*Cormack, 1964*; *Jolly, 1965*; *Seber, 1965*). Abundance ($N$) was derived from the Huggins component. Apparent annual survival reflected annual survival between primary trapping occasions and site fidelity. Temporary emigration was the probability of an individual being off the study site during a primary occasion given that individual

was not present during the previous primary occasion. Temporary immigration was the probability of being off the study area during a primary occasion given that an individual was present during the previous primary occasion. On both sites, there were 12 daily secondary occasions within each primary occasion.

We conducted our analysis in two stages. In stage one, we used a sequential modeling strategy and an Information Theory approach to model selection to develop and select the most supported model(s) using the full data set consisting of 12 trap nights. We used Akaike's Information Criterion, corrected for small sample sizes ($AIC_C$) and $AIC_C$ weights ($w_i$) to select the most-supported model in each sequential step, and each modelling step included a null model in the model selection set to evaluate model performance (*Burnham & Anderson, 2002*). We selected the model with the lowest $AIC_C$ and highest $w_i$ as our best supported model. We used the relative change in $AIC_C$ ($\Delta AIC_C$) to evaluate each model relative to the top-ranking model, and we considered models within 2 $AIC_C$ units of the top-ranking model competitive (*Burnham & Anderson, 2002*). We used simple model covariates representing the most likely sources of variation in the parameters of interest (Table 1).

The sequential modeling processes started with a global model which included a site-by-year interaction for capture, recapture, and apparent annual survival probabilities, with a random emigration model structure for emigration and immigration. We then considered seven model structures for emigration and immigration, including no movement, random movement, and Markovian movement. The no-movement model structure implies that the probability of emigration or immigration from the sites is zero. The random movement structure implies that emigration probability is the same as immigration probability for each site. Markovian movement implies that emigration or immigration are conditional on a previous state. Then, using the most-supported emigration and immigration model structure, we considered six model structures representing capture and recapture behaviors to determine if there was evidence for a behavioral response to trapping. Next, we modeled recapture probability using the most-supported model structure for emigration and immigration and the global model structure for capture probability and apparent annual survival. We then modeled capture probability using the most-supported model structure for emigration and immigration and recapture, and the global model structure for apparent annual survival. Lastly, we modeled apparent annual survival probability using the previously identified model structures for emigration and immigration, recapture, and capture probabilities.

In stage two, we fit two reduced datasets to the most supported parameter structures developed from the full 12-night data set by using the first four trapping nights only and the first eight trapping nights only. We compared mean estimates and 95% confidence intervals (hereafter 95% CI) from models fit with these reduced datasets to the mean estimates and 95% CIs from the model fit with the full dataset.

## Trap placement

We used a generalized linear model (GLM) with a Poisson error distribution to examine the effect of trap placement on the site level frequencies of flying squirrel captures. We

**Table 1** Description and sampled range of variables considered in models of capture probability ($p$), recapture probability ($c$), and apparent annual survival ($\varphi$) for Humboldt's flying squirrels captured on 16 sites in Oregon, USA.

| Covariate | Description |
| --- | --- |
| Null | An intercept only model structure. |
| Year | A year specific effect for each primary trapping occasion. |
| Time | A linear trend (1–12) from the first to the last day of trapping within a primary trapping occasion. |
| Site | A site-specific effect for each trapping location. |
| Area | A study specific effect to indicate a difference between trapping sites located within the HJ Andrews Experimental Forest, and sites located within the Siuslaw National Forest. |

used the site-, trap type- and year-specific frequencies of squirrel captures as our response variable. The fit model included one fixed effect indicating ground or tree trap placement. We assessed model fit with two methods. First, we estimated the amount of variation explained by the model using the likelihood-ratio based pseudo $R^2$ (*Cox & Snell, 1989*; *Nagelkerke, 1991*). Secondly, we compared the above model to a null model using $AIC_C$ (*Burnham & Anderson, 2002*). We estimated the group means and 95% CIs of the site- and year-specific frequencies of squirrel captures in ground and tree Tomahawk traps. We assessed the effect of ground or tree trap placement using the coverage of the group's 95% CIs, and by the amount the trap placement beta coefficient overlapped zero. We considered the effect meaningful if the groups 95% CIs did not overlap.

We performed all analyses in R (*R Core Team, 2018*). We used the RMark package version 2.2.5 to fit all Huggins Robust Design Models and assess the effects of dataset reduction on parameter estimation (*Laake, 2013*).

## RESULTS

On the HJA sites, we captured 1,076 individual flying squirrels during 62,217 trap nights. On the SIU sites, we captured 201 individual flying squirrels during 27,919 trap nights. During three weeks of live-trapping, we observed a slight increase in mean individual body mass of juvenile flying squirrels, but a slight decrease in average individual body mass of adult flying squirrels (Table 2). Juvenile flying squirrel mortality rates were higher during all three weeks of live-trapping relative to adult flying squirrels, and increased during each week. Adult flying squirrel mortality rates were much lower and varied less among weeks relative to juvenile mortality rates (Table 2).

The top-ranking temporary emigration and immigration model strongly supported a model structure with no temporary emigration or immigration (cumulative $w = 68\%$, Table S1). There was strong model selection support for a behavioral response to trapping. The top-ranking model, which estimated recapture probability separately from capture probability, received 100% of the cumulative model selection weight (Table S2). On both sites, recapture probability decreased slightly within primary occasions, from 0.31 (95% CI [0.30–0.33]) to 0.21 (95% CI [0.20–0.22]) on the HJA sites, and from 0.24 (95% CI [0.21–0.27]) to 0.15 (95% CI [0.14–0.18]) on the SIU sites where it was lower overall (Fig. 2; Table 3). Capture probability varied among years on both sites from 0.16 (95% CI [0.14–0.19]) in 2013 to 0.29 (95% CI [0.25–0.32]) in 2015 on the HJA sites and from

**Table 2 Trap week-specific average body masses (mean ± SE), mortality rates, and number of captured individual adult and juvenile Humboldt's flying squirrels captured on 16 sites during two studies in Oregon, USA.** We present mortality rates as the number of mortalities per 100 Humboldt's flying squirrel captures.

| Week | Trap nights | Juveniles | | | Adults | | |
|---|---|---|---|---|---|---|---|
| | | *n* | Mass (g) | Mortality rates | *n* | Mass (g) | Mortality rates |
| 1 | 1–4 | 318 | 89.52 ± 0.92 | 1.22 | 549 | 128.81 ± 0.49 | 0.72 |
| 2 | 5–8 | 408 | 93.46 ± 0.82 | 1.39 | 574 | 126.23 ± 0.50 | 0.57 |
| 3 | 9–12 | 446 | 95.96 ± 0.73 | 2.23 | 603 | 124.96 ± 0.51 | 0.65 |

0.05 (95% CI [0.03–0.08]) in 2014 to 0.07 (95% CI [0.05–0.11]) in 2015 on the SIU sites where similar to recapture probability it was lower overall (Fig. 3; Table 4). Capture probability estimates were unidentifiable during 2011 on the HJA sites while using the 4- and 8-night data sets. Apparent annual survival was variable among years on the HJA sites where it decreased from 56% (95% CI [44%–67%]) during 2011–2012 interval to 33% (95% CI [27%–39%]) during the 2015–2016 interval, but was not variable on the SIU sites where it ranged from 40% (95% CI [28%–54%]) during 2014–2015 and 40% (95% CI [27%–53%]; Fig. 4). Similar to capture probability, apparent annual survival estimates were unidentifiable during the 2011–2012 interval on the HJA sites while using the 4-night data set. Site- and year-specific abundance estimates on the HJA sites ranged from 6.2 squirrels (95% CI [4.4–15.9]) in 2011 to 63.4 squirrels (95% CI [59.2–72.9]; Fig. 5), whereas abundance estimates on the SIU sites ranged from 2.1 squirrels (95% CI [1.1–9.6]) to 54 squirrels (95% CI [38.9–89.2]), both during 2014 (Fig. 5).

On the HJA sites, we observed nearly no change in recapture probability estimates between 8 nights and 12 nights (mean 1.0006-fold, range = 0.97–1.02), but a mean 1.09-fold (range = 1.01–1.16) decrease in recapture probability for 4 nights as compared to 12 nights. On the SIU sites, we observed a mean 1.22-fold (range = 1.19–1.25) decrease in recapture probability when using 8 nights, and a mean 2.03-fold decrease (range = 1.86–2.19) when using 4 nights. In addition, when using 4 nights, the direction of the time effect differed from 12 nights (Fig. 2). Capture probability was overestimated when trapping occasions were reduced; however, the effect was much stronger when using 4 nights than 8 nights (Fig. 3). Compared to 12-night estimates, we observed a mean 1.13-fold (range = 1.05–1.20) decrease in capture probability when using only 8 nights, and a mean 4.61-fold (range = 4.17–5.19) increase on the SIU sites when using only 4 nights, whereas on the HJA sites we observed a mean 1.24-fold increase (range = 0.96–1.36) when using 8 nights, and a mean 1.8-fold increase (range = 1.63–2.02) when using 4 nights. Apparent annual survival estimates were stable across subsets in comparison, except for 4 nights on the SIU sites, where we observed a mean 2.20-fold (range = 1.93–2.45) decrease relative to other estimates (Fig. 4). Abundance was underestimated relative to the full dataset when using both 4 and 8 nights, but the underestimation was most extreme when using 4 nights (Fig. 5). On the SIU sites, we observed a mean 1.06-fold (range = 0.63–2.91) decrease in abundance when using 8 nights and a mean 3.85-fold (range = 1.31–8.57) decrease when using 4 nights relative to 12-night estimates. On the HJA sites, relative to 12 nights, we
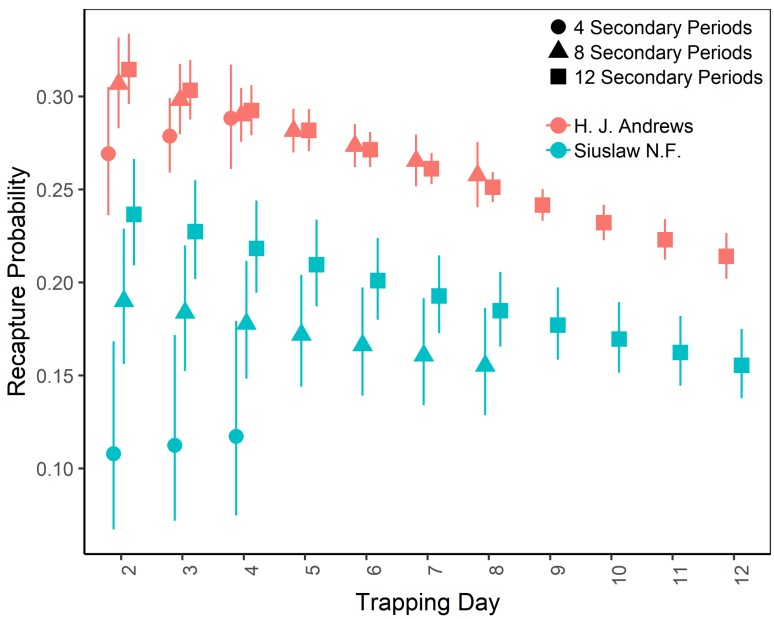

**Figure 2** **Recapture probabilities of Humboldt's flying squirrels estimated using Huggins robust design models and live-capture data collected in the HJ Andrews Experimental Forest and the Siuslaw National Forest in western Oregon.** Estimates from three nested subsets of data for each study area with vertical bars indicated the 95% confidence intervals.

observed a mean 1.09-fold (range = 0.88–1.42) decrease in abundance when using 8 nights, and a mean 1.38-fold (range = 1.00–2.38) decrease when using 4 nights.

Trap placement in trees strongly increased ($\beta_{\text{TreeTrap}}$: 0.70 95% CI [0.64–0.76]) the frequency of squirrel captures and resulted in higher site- and year-specific squirrel capture frequencies (mean frequency = 47.68 95% CI [46.17–49.26]) relative to traps placed on the ground (mean frequency = 23.68 95% CI [22.61–24.80]) for both study sites. In addition, model selection results strongly supported the inclusion of a trap type covariate relative to the null model. Likelihood-ratio based pseudo $R^2$ ($R^2 = 0.98$) for the GLM model explained the variance of site-, trap type-, and year-specific frequencies of flying squirrel captures well.

## DISCUSSION

We assessed the effect of reducing trap nights on the estimation of apparent annual survival, capture probability, recapture probability, and abundance of flying squirrels using Huggins robust design capture-recapture models. As predicted, we were able to obtain reliable and precise estimates for the parameters of interest with less than three weeks of live-trapping data after the first year of trapping. During the first year of trapping on the HJA sites, capture probability and apparent annual survival estimates were unidentifiable while using the reduced datasets. We found that reducing the number of trap nights from 12 to eight had little impact on parameter estimates or estimates of precision, whereas further reducing the number of trap nights to four resulted in substantial estimate apparent bias

**Table 3  Models used to determine the most parsimonious recapture probability ($c$), capture probability ($p$), and apparent annual survival ($\varphi$) model structures for Humboldt's flying squirrels captured on 16 sites during 2 studies in Oregon, USA.** We present model structure change in Akaike's Information Criterion adjusted for sample size ($AIC_C$) from the top-ranking model ($\Delta AIC_C$), $AIC_C$ weight of evidence ($w$), and the number of parameters ($K$).

| Parameter | Model | $\Delta AIC_C$ | $w$ | $K$ |
| --- | --- | --- | --- | --- |
| $c^a$ | Time + Area | 0.00 | 0.94 | 94 |
| | Year + Area | 5.44 | 0.06 | 98 |
| | Time | 30.09 | 0.00 | 93 |
| | Area | 54.05 | 0.00 | 93 |
| | Year | 62.71 | 0.00 | 97 |
| | Null | 91.13 | 0.00 | 92 |
| $p^b$ | Year + Area | 0.00 | 0.82 | 69 |
| | Time + Area | 3.03 | 0.18 | 65 |
| | Area | 27.43 | 0.00 | 64 |
| | Time | 82.70 | 0.00 | 64 |
| | Year | 106.46 | 0.00 | 68 |
| | Null | 110.68 | 0.00 | 63 |
| $\varphi^c$ | Year + Area | 0.00 | 0.86 | 12 |
| | Year | 3.65 | 0.14 | 11 |
| | Area | 11.32 | 0.00 | 8 |
| | Null | 12.18 | 0.00 | 7 |

Notes.
[a] Model structures for initial capture ($p$) probability were held to a site by trapping day model structure (Site * Time), and model structures for apparent annual survival ($\varphi$) were held to a site by year model structure (Site * Year), while model structures for emigration and immigration were fixed to zero.

[b] Model structures for apparent annual survival ($\varphi$) were held to a site by year model structure (Site * Year), while emigration and immigration model structure was fixed to zero, and recapture probability ($c$) was held to an additive time and area (Time + Area) model structure.

[c] Model structures for emigration and immigration model structure was fixed to zero, recapture probability ($c$) was held to an additive time and area (Time + Area) model structure, and capture probability ($p$) was held to an additive year and area (Year + Area) model structure.

and decreased precision. As expected, the estimates of parameter precision were inversely associated with the number of trapping nights. In most cases the point estimates and 95% confidence intervals for the 8- and 12-night estimates overlapped. However, further reducing the dataset to 4 nights resulted in variable point estimates with 95% confidence intervals that did not overlap with the 12-night estimates. In addition, the effect direction of the trapping occasion covariate differed between the 4-night estimate and both the 8- and 12-night estimates. In this case recapture probability increased across the first four nights when using 4 nights, while recapture probability decreased during the same nights with both the 8 and 12- nights. In all cases, recapture probabilities were underestimated, capture probabilities were overestimated, and abundances were underestimated as the number of trap nights decreased. Apparent annual survival estimates were most robust to reduced numbers of trap nights; only in some years were the point estimates and precision estimates affected by reducing the number of trap nights.

Fall abundances were variable among years and were generally higher on the HJA sites relative to SIU sites. Similarly, capture probability was substantially higher on HJA sites

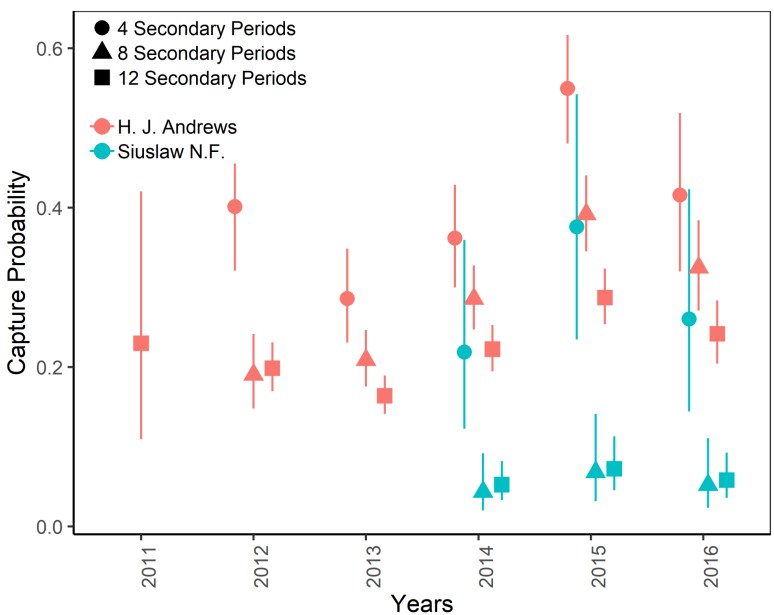

**Figure 3** Capture probabilities of Humboldt's flying squirrels estimated using Huggins robust design models and live-capture data collected in the HJ Andrews Experimental Forest and the Siuslaw National Forest in western Oregon. Estimates from 3 nested subsets of data for each study area with vertical bars indicated the 95% confidence intervals.

**Table 4** Logit scale estimates and 95% confidence intervals for covariate effects from the top-ranking Huggins robust design model for Humboldt's flying squirrels captured on 16 sites during 2 studies in Oregon, USA.

| Parameter | Covariate | Estimate | 95% CI | |
|---|---|---|---|---|
| | | | Lower | Upper |
| $c$ | Intercept | −0.78 | −0.87 | −0.69 |
| | Time | −0.05 | −0.07 | −0.04 |
| | Area SIU | −0.39 | −0.53 | −0.25 |
| $p$ | Intercept | −1.14 | −1.36 | −0.92 |
| | Year 2011 | −0.07 | −0.98 | 0.85 |
| | Year 2012 | −0.25 | −0.54 | 0.04 |
| | Year 2013 | −0.49 | −0.76 | −0.21 |
| | Year 2014 | −0.11 | −0.38 | 0.16 |
| | Year 2015 | 0.23 | −0.04 | 0.50 |
| | Area SIU | −1.64 | −2.12 | −1.16 |
| $\varphi$ | Intercept | −0.72 | −0.99 | −0.44 |
| | Year 2011–2012 | 0.96 | 0.42 | 1.50 |
| | Year 2012–2013 | 0.58 | 0.21 | 0.96 |
| | Year 2013–2014 | 0.18 | −0.17 | 0.52 |
| | Year 2014–2015 | 0.02 | −0.33 | 0.36 |
| | Area SIU | 0.29 | −0.27 | 0.85 |

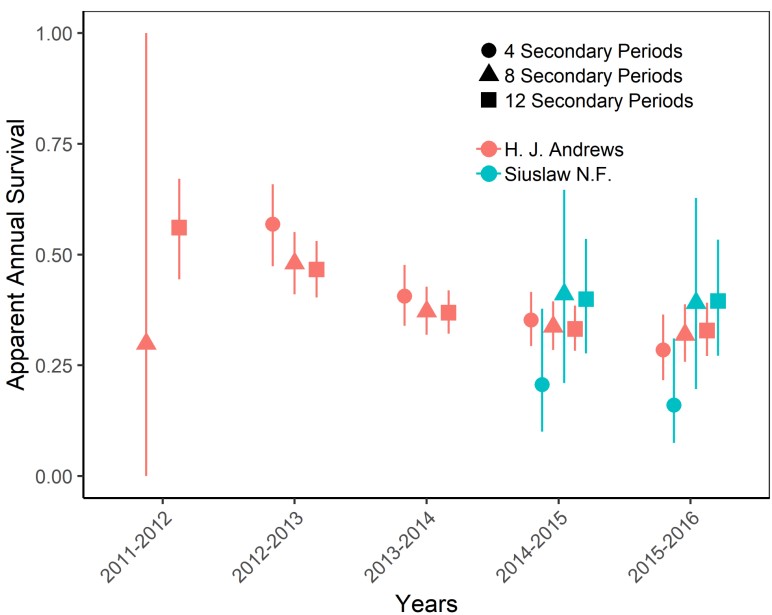

**Figure 4** **Apparent annual survival of Humboldt's flying squirrels estimated using Huggins robust design models and live-capture data collected in the HJ Andrews Experimental Forest and the Siuslaw National Forest in western Oregon.** Estimates from 3 nested subsets of data for each study area with vertical bars indicated the 95% confidence intervals.

relative to SIU sites. The abundance estimates presented here are similar to those reported by *Weldy et al. (2019)* which were obtained using HJA data. Apparent annual survival was similar on both the HJA and SIU sites. We used robust design temporal symmetry models and observed slightly more temporal variation in apparent annual survival than (MJ Weldy, CW Epps, DB Lesmeister, T Manning, and E Forsman, 2019, unpublished data). But, overall temporal variation in apparent annual survival was small relative to the temporal variation in other parameter estimates. In addition, estimates from both the HJA and SIU sites were generally similar and were within the range of other recent estimates of apparent annual survival for Humboldt's flying squirrels and northern flying squirrels (0.32–0.68 as reported by *Ransome & Sullivan, 2002*; *Gomez, Anthony & Hayes, 2005*; *Lehmkuhl et al., 2006*).

A reduction in trapping effort to obtain abundance estimates would help reduce overall trapping costs and help minimize effects on the health and well-being of small mammals sampled during live-trapping studies. Labor expenses can be the greatest cost associated with trapping, after initial investment in traps and related equipment. Reducing the number of trapping days by one-third would, in turn, reduce the per-site labor costs by nearly the same amount. Lower labor costs could influence future studies in two ways: (1) researchers could then increase spatial or temporal replications within a study, and (2) the range of covariates sampled in association with any trapping effort could be expanded.

We observed consistent or increasing mortality rates across three weeks of live-trapping. Reducing trapping effort to two weeks would shorten the trap exposure time of animals

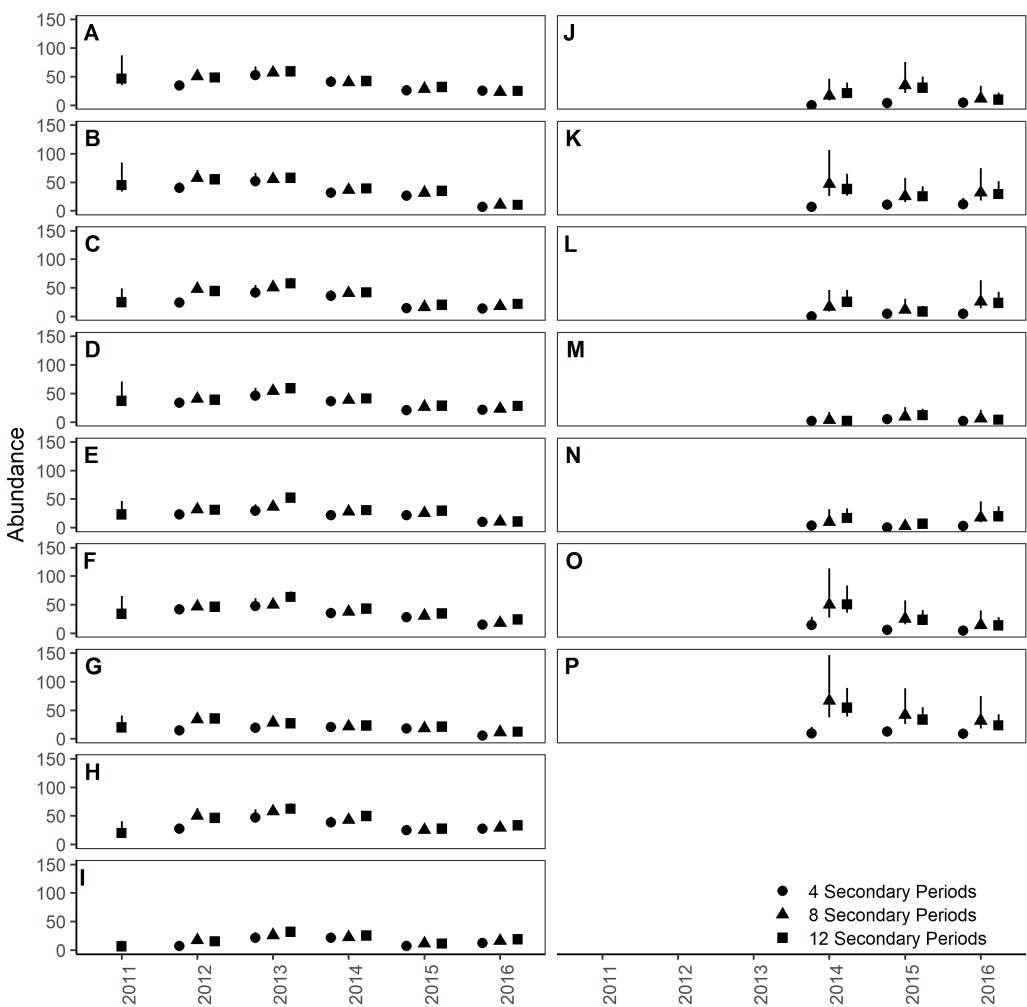

**Figure 5** **Site- and year-specific abundances of Humboldt's flying squirrels estimated using Huggins robust design models and live-capture data collected on 7.84 ha sites in the HJ Andrews Experimental Forest and the Siuslaw National Forest in western Oregon.** Estimates from three nested subsets of data for the HJ Andrews (A–I) and Siuslaw National Forest (J–P) with vertical bars indicate the 95% confidence intervals.

during a given trapping session, which in turn would reduce the overall number of mortalities. It would also help reduce injuries associated with trapping (e.g., skin abrasions, hypothermia). This may be especially true for "trap-happy" individuals that spend much of each week repeatedly captured in traps and for incidental species that are prone to stress myopathy (e.g., Douglas' squirrels, *Tamiasciurus douglasii*). We used individual body mass as a proxy for body condition, but recognize the limitations that body mass has on representing animal fitness. For example, average juvenile body mass increased during each week of trapping, but it is unclear if trapping slowed increases in average juvenile body relative to uncaptured juveniles, potentially affecting winter survival (Table 2). Likewise, we observed a decline in adult weights over time but without knowing whether this affected fat reserves or just represented normal fluctuations in body mass, we would urge

caution in interpreting these results. The stomach and contents of adult flying squirrels can represent >1/3 of total body mass (*Villa et al., 1999*). However, the consequences of reducing effort likely depend on the overall project goals, and careful consideration of the target population, trapping sites, and study goal features is important to study design.

We prebaited the HJA traps, but not the SIU traps. This difference could have influenced our abundance estimates, but we believe habitat quality was more likely to influence estimates than pre-baiting. If prebaiting had a strong effect we would have expected abundance and capture probability estimates from the 4- and 8-night subsets to perform relatively worse on the SIU compared to 12 nights. The 4-night subsets did perform substantially worse than both the 8-night subset and the 12-night dataset, but there was no strong evidence that the 8-night subset performed worse than the 12-night dataset. These results suggest prebaiting might have strong effect in situations where limited resources or experimental design impose short (<5 night) trapping occasions. Yet, the effect appears diminished during relatively longer (>8 night) trapping occasions. We observed much higher recapture and capture probabilities on the HJA sites relative to the SIU sites. We are, however, uncertain if this observation is an effect of prebaiting or if the effect is a result of differences in landscape context or habitat suitability. Previous studies have explored the effects of prebaiting small mammal traps and have found mixed results, indicating variable effects of prebaiting on trapping success. For example, *Chitty & Kempson (1949)* were among the first to suggest that prebaiting could familiarize species to newly placed traps before sampling began. *Gurnell (1980)* suggested that prebaiting was only effective if the trapping period was short. More recently, *Edalgo & Anderson (2007)* reported that prebaiting did not enhance trapping success in prebaited traps relative to traps that were not prebaited.

Contrary to our prediction, flying squirrels were captured more frequently in tree traps than ground traps. Our findings support the findings of *Carey, Biswell & Witt (1991)*, and suggest that tree traps do improve the frequency of flying squirrel captures and are likely important for targeting flying squirrels despite longer setup and check times. However, we are uncertain if the mechanism behind this observation reflects a preference for tree traps by flying squirrels or a decrease in ground trap availability resulting from daytime captures of other species, especially Townsend's chipmunks (*Neotamias townsendii*). *Risch & Brady (1996)* found that southern flying squirrels (*Glaucomys volans*) were captured more frequently in traps placed on tree boles at >4.5 m when compared to traps placed at approximately 2 m, so the height at which traps are placed in trees may also influence squirrel captures. We are unaware of any studies in the PNW that have tested variation in trap tree height for either Humboldt's flying squirrels or northern flying squirrels and we suggest that further study on this topic may be warranted.

## CONCLUSIONS

Small mammal abundances and vital rates will likely continue to be of research interest in the PNW and beyond. The use of analytical methods accounting for imperfect detection has increased (*Kellner & Swihart, 2014*) and methods for analytically incorporating trapping

methodology will continue to change. As a result, periodic reviews of trapping methods can be a fruitful exercise that may result in reducing project costs and minimizing the effects of trapping projects on the health and well-being of both target and non-target species. We found that estimates of abundance, apparent annual survival, and capture and recapture probabilities for Humboldt's flying squirrels based on eight secondary occasions were largely equivalent to those based on 12, but estimates from only four secondary occasions were biased relative to those based on 12 and insufficient to achieve precision. We also found support for use of tree traps to increase capture frequency. Our results provide a framework for methodological review that could be useful for live trapping studies involving Humboldt's flying squirrels and could be extended to other species or other geographic locations.

## ACKNOWLEDGEMENTS

We would like to thank Tom Manning and Mark Linnell for their insights and logistical support. We would also like to thank the numerous technicians and volunteers who assisted with collection of the trapping data: S Adams, NB Alexander, DA Arnold, AM Bartelt, A Bies, SC Bishir, NA Bromen, LL Carver, D Clayton, MJ Cokeley, J Dunn, CR Gray, E Halcomb, L Hodnett, LK Howard, AC Hsiung, CJ Hutton, DK Jacobsma, PC Kannor, BE Kerfoot, L Kerstetter, A Kupar, M Lovelace, M Massie, TJ Mayer, B Nahorney, E Oja, HM Oswald, SM Pack, B Peterson, T Phillips, KA Ray, M Scoggins, J Soy, G Shank, DC Tange, C Ward, SE Ward, and JM Winiarski. The findings and conclusions in this publication are those of the authors and should not be construed to represent any official U.S. Department of Agriculture or U.S. Government determination or policy. The use of trade or firm names in this publication is for reader information and does not imply endorsement by the U.S. Government of any product or service.

### Funding
Small-mammal trapping and analysis was supported by the U.S.D.A. Forest Service, Pacific Northwest Research Station. The funders had no role in study design, data collection and analysis, decision to publish, or preparation of the manuscript.

### Grant Disclosures
The following grant information was disclosed by the authors:
U.S.D.A. Forest Service, Pacific Northwest Research Station.

### Competing Interests
The authors declare there are no competing interests.

### Author Contributions
- Matthew J. Weldy conceived and designed the experiments, performed the experiments, analyzed the data, contributed reagents/materials/analysis tools, prepared figures and/or tables, authored or reviewed drafts of the paper, approved the final draft.

- Todd M. Wilson and Clinton W. Epps conceived and designed the experiments, performed the experiments, contributed reagents/materials/analysis tools, authored or reviewed drafts of the paper, approved the final draft.
- Damon B. Lesmeister performed the experiments, contributed reagents/materials/analysis tools, authored or reviewed drafts of the paper, approved the final draft.

## Animal Ethics

The following information was supplied relating to ethical approvals (i.e., approving body and any reference numbers):

The HJA live-trapping protocols were approved by Oregon State University's Institutional Animal Care and Use Committee (ACUP: #4191 (2011–2013), #4590 (2014–2016)); and the SIU live-trapping protocols were approved by the USDA Forest Service Starkey Institute for Animal Care and Use Committee (USFS: #92-F-0004). In addition, both live-trapping protocols were consistent with the American Society of Mammalogists guidelines for the use of wild mammals in research and education Sikes (2016).

## Field Study Permissions

The following information was supplied relating to field study approvals (i.e., approving body and any reference numbers):

Field work for this project was collected under Oregon Department of Fish and Wildlife scientific take permits from 2011 to 2016 (STP #s 122-11, 109-12, 039-13, 052-14, 118-14, 047-15, 124-15, 058-16, and 094-16). This work conducted on Forest Service lands with approval from the Siuslaw National Forest, Willamette National Forest, and the HJ Andrews Forest research group.

## Data Availability

Mark-recapture data, trap-type specific capture frequency data and R code to analyze the mark-recapture data and capture frequency data are available as Supplemental Files.

## Supplemental Information

Supplemental information for this article can be found online at http://dx.doi.org/10.7717/peerj.7783#supplemental-information.

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
