# Peer review of "Effects of trapping effort and trap placement on estimating abundance of Humboldt’s flying squirrels"

_PeerJ, doi:10.7717/peerj.7783_

## Round 0.1 · original submission · Minor Revisions

This is a very nice contribution and informs methods and analysis related to estimations of abundance which are important and timely. The reviewers have suggested some minor revisions as well as a few additions which we feel would improve the manuscript. I think that that these minor revisions will improve the manuscript and should be quite easy to take on. We look forward to receiving a revised version and thank you for your contribution.

Reviewer 1 ·

Basic reporting

No comment

Experimental design

No comment.

Validity of the findings

No comment.

Additional comments

This is a nice short paper that looks at trap placement, and how many trap nights are required to get reasonable population parameters for Humboldt’s flying squirrel: the answer probably being placing traps on tree trunks for 8 nights. The authors estimate abundance, capture probability, and recapture probability over 4, 8, and 12 trap nights using two datasets spanning 6 and 4 years from several sites in HJA and SIU forests. Trapping method was mark-recapture using Tomahawk live traps, and conducted in accordance to proper animal care and handling procedures. Data was analyzed using Huggins robust design modeled in RMARK, which is an appropriate and well accepted method for deriving population parameters.

In short, this is a well written paper, with clear methods, rigorous analysis and results reporting, and sound description of conclusions. The trapping methodology was carried out according to proper animal care procedures. The study was granted appropriate permits, and no undue stress was issued to the animals.



I do not have many suggested edits, but I would like to see an addition in the Results and Discussion exploring how trapping for 4-, 8-, and 12-nights affected animal condition or mortality. The authors mention that reducing trap nights from 8 to 12 reduces stress on animals, but the authors do not explore it (line 332-344). This addition would add valuable and pertinent information to their article, and should not be particularly onerous. Adding animal condition information would be especially important considering the authors describe cases where researchers have increased trapping duration from 8 to 12 nights to improve capture numbers in low density populations. What is the actual cost for animals trapped over 12 nights as opposed to 8? Is there an increase in mortality, or a decline in apparent condition of the animals?


Line 114: I would like to see a small map of the area added in

Line 117-120: SIU elevations are reported twice, and are different

Line 332-344: There is some speculation here. I would like to see this section expanded with actual data to further explore how animal condition changes over 4-, 8-, and 12-trap nights.

Line 348-350: I agree with the authors’ conclusion that prebaiting does not seem to impact trapping periods of 8 or 12 days. However, the authors do not mention the effect of prebaiting on the 4-night period, where prebaiting would arguably have the greatest impact, and which does appear to be different. I would like to see this paragraph expanded to include the impact of prebaiting on the 4-night trapping session.

·

Basic reporting

The basic reporting in this manuscript is well done and conforms to Peer J standards and raw data were provided in supplemental files

Experimental design

The manuscript appears to represent original primary research and is within scope of the journal. The basic research questions are well defined, relevant and meaningful and It is stated how the research fills an identified knowledge gap.
It appears that rigorous investigation was performed to a high technical and ethical standard. The field and analytical methods are described in sufficient detail to replicate.

Validity of the findings

Based on my examination, the data appear to be robust and statistically sound.

Additional comments

In this paper, the authors examine the efficiency of various live trapping strategies (ie, trap placement and successive nights trapped) on a small mammal, Humboldt’s flying squirrel. They evaluated how bias and precision of estimates for apparent annual survival, abundance, capture probability, and recapture probability of Humboldt’s flying squirrels varied with the number of secondary trapping occasions.

The authors used a sequential modeling procedure to analyze their trapping data and found that reducing the length of primary trapping occasions from 12 to 8 nights had very little impact on parameter estimates, but further reducing the length of primary trapping occasions to 4 nights caused substantial bias in parameter estimates and decreased precision. Thus, this study suggests that 8 nights of trapping is sufficient to effectively sample Humboldt’s flying squirrels, reducing the need for extra nights which would likely cause extra expense and may cause unnecessary stress to animals. The authors also found that attaching Tomahawk traps to trees increased the site- and year-specific capture frequency of flying squirrels. This is consistent with my own personal experience trapping for these squirrels.

Overall, this is a solid, well done and well written paper. It provides useful information for field biologists working on Humboldt’s flying squirrels, and provides a road map for evaluating optimal trapping strategies for other species. The figures are clear and useful and the methods and results are well described.

This is an interesting, well done paper that provides useful information on study design for live trapping Humboldt’s flying squirrels, and is a good example of how to determine an efficient live trapping regime for small mammals in general. I liked the sequential modeling approach the authors used to analyze their data; it is logical and seems statistically sound to me. The authors also confirmed that placing some traps on tree trunks increased trapping efficiency. This is a common practice, but it takes extra time in the field to attach live traps to trees, so it is nice to know that the extra work seems to have a positive return.

One suggestion I have is to add a citation for Humboldt’s flying squirrel in the introduction around line 70. This species was just described in 2017, and many biologists might still not be familiar with it and it’s Latin name, Glaucomys oregonensis. Something like “Two species of flying squirrels, the recently described Glaucomys oregonensis (Arbogast et al 2017) and G. sabrinus, have been of interest because they serve as prey for the federally threatened northern spotted owl...”

I also think the Figure Legends could be tightened up a little. For example, for Fig 1, consider “Recapture probabilities of Humboldt’s flying squirrels in the H. J. Andrews Experimental Forest and the Siuslaw National Forest in western Oregon. Estimates from 3 nested subsets of data are shown for each study area with vertical bars indicated the 95% confidence intervals.”

·

Basic reporting

Basic reporting is reasonable.

Experimental design

Design is sound and fundamental questions are well defined.

Validity of the findings

Findings are well-supported and reasonable.

Additional comments

The authors use capture-recapture data on flying squirrels collected during a series of 12-occasion surveys from two areas to explore the consequences of reducing effort to 8 and 4 days on parameter estimates and to evaluate the relative effectiveness of traps placed in two different locations. Although I offer a number of comments and suggestions below, I think the paper is solid and well written. My main criticism is that the focus is quite narrow for a paper exploring design alternatives. Consequently, my principal suggestion focuses on broadening the scope of the work to increase the relevance of the paper to other species and to a wider range of sampling designs.

One issue with using only field data to explore design options is that we do not know the true value of parameters, which eliminates the possibility of exploring bias (see below) and makes exploring precision complex for several reasons, one of which it becomes conditional on the choice of model for analysis. Although I appreciate the realism we get from field data (and I am an empiricist), simulations provide clearer insights on the effects of altering design options. Therefore, my primary suggestion is to add some simulations to the paper. That will require a modest amount of effort, but what you learn through that exercise will complement the results from field data and strengthen the paper in multiple ways, one of which is allowing you to delineate more carefully the influence of effort on bias and precision.

Specifically, exploring a range of plausible values for the key parameters under a series of scenarios would allow those interested in other species to benefit from your work and allow you to view flying squirrels as a case study. Because capture and recapture probabilities are quite low for flying squirrels (especially at one of your sites), your work only has relevance to a narrow subset of similar species and similar sampling designs. I appreciate the practicality of 4, 8, and 12 occasions in your specific study, but exploring a wider range (e.g., 4-12) would help to illustrate and inform a range of other possible design alternatives. Same holds for exploring a range of different values for the target demographic parameters and detection-related parameters. You would not need to explore all that many values for each parameter to inform choices relevant to a wider range of species and sampling designs. In fact, in addition to the values you report for flying squirrels, I’d inform the ranges of values to explore based on empirical work from other small mammals.

Below, I’ve provide a number of line-by-line comments. Although some are substantive, I see them as relatively minor compared to broadening the scope.

Line 20. I’d be cautious about using the word bias to represent deviation from estimates based on 12-occasion surveys. Strictly, bias is deviation between the expected value of an estimate and the true value of the parameter, which you don’t know (but would with simulations). I don’t think it’s unreasonable to assume the 12-occasion estimates are ‘better’ than those based on fewer sampling occasions, but you have no real way to gauge that. So I think the quantity is useful to report, but I would call it something different (maybe apparent bias?).

Line 41 and elsewhere. You use abundance and population size interchangeably; probably best to be consistent (I’d vote for abundance).

Line 97. Smaller than needed to achieve what objective? These sorts of statements only make sense for specific objectives.

Line 160. Sticking with one approach eliminates model bias, so I don’t mind that you only considered only Huggins’ models. But I think you should explain why you made that choice and whether you explored other approaches. To be clear, I don’t think you need to include information on that initial effort (assuming there was one), but I was curious how and why you chose the one you did.

Lines 179-210. Unless I missed it, you do not discuss the criteria you used to choose among models (I assume you established an arbitrary value of AICc) or how you dealt with competing models.

Line 197. Replace ‘if the data supported the modelling of’ with ‘if there was evidence for’

Line 198. Considering a behavioral response to trapping seems fine, but I wondered why you didn’t also pursue heterogeneity (and perhaps b + h) as it emerges as a reasonable model for many species.

Line 215. If site and year were modeled as random effects (which I think you’re implying), I’d mention that explicitly.

Line 216. It’s no big deal, but I don’t understand the value of R-squared when the question revolves around a single parameter. In the results (lines 290-296), R-squared gets the lion’s share of attention and the actual result is relegated to the end. Therefore, I’d suggest moving the result to the front of the paragraph and offering R-squared parenthetically (if at all).

Evaluating the question of model fit is a lot more valuable when the goal is parameter estimation, but you did not include it there and probably should (i.e., goodness-of-fit tests for your top models).

Line 239 and throughout this section. You should include units for each estimate you report otherwise readers might think they are direct probabilities, when each beta is on the scale of the link function.

Line 245. Something seems off here – perhaps an aberrant ‘to 0.53’? More importantly, this entire section is challenging to work through, so you might consider eliminating contrasts such as this when they are based solely on point estimates (i.e., the CIs largely overlap, so I am not sure claims of systematic patterns are especially compelling).

Line 246-7. Seems awkward to use both a model-selection approach (I suspect by ‘some support’ you mean that one or more of the competing models included a site effect) and a frequentist approach (because they are based on an alpha level, CIs are frequentist measures). Seems cleanest to me to simply go with whether the CI for beta overlaps zero as you do elsewhere.

Line 249 and throughout this section. You sometimes report probabilities as percentages; probably best to stick to probabilities.

Lines 262-266. I think it would be better to convert these to a per hectare basis rather than on the scale of the plot.

Line 273. Delete the word ‘parameter’

Lines 290-296. See my comment offered for line 216.

Lines 307-309. I’m not sure I see the contrast implied by the ‘but’ in the middle of the sentence. We don’t expect bias to change systematically with sample size, just precision (as you note), so CIs should always overlap.

Lines 311-314. This all seems likely to be an artifact of low capture probabilities, which is true for many of the patterns you report. Here, based on Figure 2, CIs around the point estimates for the 4-occasion surveys seem to include a neutral or even negative beta, so it doesn’t seem that the change in sign is especially meaningful. The issue is likely an artifact of so few recaptures for a species with low detection rates and even lower recapture rates. That seems to be the important point – you need sufficient data to get good estimates.

Lines 332-344. Of course, your goal is to explore the consequences of reducing effort, but as I mentioned earlier, whether than is a reasonable trade-off will depend entirely on the overarching objective. For long-term work, optimal sampling designs will always be different than those for short-term work. I think it would be worthwhile to readers to explain how design is always a series of trade-offs and that the ideal design depends largely on the features of the target population and the target objective.

Figure 1. Unlike Figure 2, each set of related values illustrated here are model-based prediction based on a constant beta, so it could be illustrated more simply by plotting the betas and their CIs. More importantly, as the paper now stands, the focus is effectively on how precision of estimates changes based on different subsets of data. Therefore, I’d think it would be more effective to illustrate directly how precision (say SE or CI width) of parameter estimates changes as the number of sampling occasions changes, rather than illustrating model effects, which is not the primary focus (at least according to the title).

Also, you should add the word ‘Estimated' to the start of the legend of this and related figures and indicate the model you’re using to estimate the values displayed.

Table 1. Given that temporary immigration and emigration are model parameters, you should at least mention that you held them constant in the table heading. More importantly, I am curious as to why you chose to include site as a covariate rather than a few habitat covariates (e.g., tree density, stand age, etc.). Sites is okay, but requires many more parameters (unless fit as a random effect) and carries no ecological information, so a few key habitat might be more interesting and more parsimonious.

Table 2. I don’t follow the information in the footnote – specifically, I am not sure what you mean by ‘held to site.’ If you mean identifying sites as structural units in the analysis, wouldn’t adding a simple site effect allow for additive differences among sites (i.e., there’s no need for interactions)? Perhaps I have that wrong, but either way, it would be good to clarify this section and even include a section in the methods if it’s complicated. Lastly, I don’t think the AICc column is useful.

I hope my comments are of some value to you.

Best wishes,
Bob Steidl

---

## Round 0.2 · accepted · Accept

Thank you for addressing the comments and suggestions from reviewers. You have addressed any and all concerns and we are happy to move this forward to publication.